# The Importance of Gut Microbiota on Choline Metabolism in Neurodegenerative Diseases

**DOI:** 10.3390/biom14111345

**Published:** 2024-10-23

**Authors:** Majid Eslami, Farnaz Alibabaei, Ali Babaeizad, Seyedeh Zahra Banihashemian, Mahdi Mazandarani, Aref Hoseini, Mohammad Ramezankhah, Valentyn Oksenych, Bahman Yousefi

**Affiliations:** 1Department of Bacteriology and Virology, Semnan University of Medical Sciences, Semnan 35134, Iran; m.eslami@semums.ac.ir; 2Student Research Committee, School of Medicine, Semnan University of Medical Sciences, Semnan 35134, Iran; farnazbabaei@gmail.com; 3School of Medicine, Semnan University of Medical Sciences, Semnan 35134, Iran; babaeizad.ali@gmail.com (A.B.); seyedehzahra.banihashemian@gmail.com (S.Z.B.); 4Endocrinology and Metabolism Research Center, Faculty of Medicine, Tehran University of Medical Sciences, Tehran 11369, Iran; mahdi_mazandarani@yahoo.com; 5Student Research Committee, School of Medicine, Mazandaran University of Medical Sciences, Sari 49414, Iran; arefhoseini148@gmail.com; 6Student Research Committee, Faculty of Medicine, Babol University of Medical Sciences, Babol 47134, Iran; mohamadramezankhah@gmail.com; 7Department of Clinical Science, University of Bergen, 5020 Bergen, Norway; 8Cancer Research Center, Faculty of Medicine, Semnan University of Medical Sciences, Semnan 35134, Iran

**Keywords:** choline, metabolism, microbiota, neurodegenerative diseases, TMAO

## Abstract

The gut microbiota is a complex ecosystem that influences digestion, immune response, metabolism, and has been linked to health and well-being. Choline is essential for neurotransmitters, lipid transport, cell-membrane signaling, methyl-group metabolism and is believed to have neuroprotective properties. It is found in two forms, water-soluble and lipid-soluble, and its metabolism is different. Long-term choline deficiency is associated with many diseases, and supplements are prescribed for improved health. Choline supplements can improve cognitive function in adults but not significantly. Choline is a precursor of phospholipids and an acetylcholine neurotransmitter precursor and can be generated de novo from phosphatidylcholine via phosphatidylethanolamine-N-methyltransferase and choline oxidase. Choline supplementation has been found to have a beneficial effect on patients with neurodegenerative diseases, such as Alzheimer’s disease (AD), by increasing amyloid-β, thioflavin S, and tau hyper-phosphorylation. Choline supplementation has been shown to reduce amyloid-plaque load and develop spatial memory in an APP/PS1 mice model of AD. Choline is necessary for normative and improved function of brain pathways and can reduce amyloid-β deposition and microgliosis. Clinical research suggests that early neurodegenerative diseases (NDs) can benefit from a combination of choline supplements and the drugs currently used to treat NDs in order to improve memory performance and synaptic functioning.

## 1. Gut Microbiota

The human gut is home to a large community of microorganisms known as the gut microbiota. These microbes live symbiotically in intestines and form one of the most complex and diverse microbial ecosystems on the planet. This unique ecosystem serves a variety of functions, ranging from aiding digestion, to helping modulate immune responses, to keeping harmful bacteria and viruses in check. The microbiome contains an estimated 100 trillion bacterial cells, which is about 10 times greater than the number of human cells in the body [1].

It can be divided into two main types: symbiotic (beneficial) bacteria and pathogenic (disease-causing) microbes. These two different types of bacteria interact with each other to form a complex ecosystem inside the intestine that influences digestion, immune response, and metabolism. In recent years, research has shown that the composition of gut microbiota plays an integral role in health and well-being. Studies have linked imbalances in the gut microbiome to a variety of disorders, such as inflammation and metabolic diseases. On the other hand, having a healthy and diverse gut microbiome has been linked to improved digestive health, better mental health outcomes, and a reduced risk for certain diseases [2].

Diet is one important factor that influences the composition of the gut microbiota. Fiber-rich diets have been shown to encourage beneficial microbes, while processed foods may promote unhealthy bacterial populations [3]. Researchers are also looking into other factors that can influence the microbiome, such as antibiotics or stress levels. Scientists hope that being able to identify distinct patterns associated with different states of health will inform decisions surrounding nutrition and lifestyle choices in order to maintain or restore balance within guts [4]. Gut microbiota play a much bigger role in overall health than most people know. Beyond aiding digestion and aiding absorption of nutrients, research has uncovered numerous ways that gut microbiota can affect your wellbeing [5].

Good mental health: Research studies have linked mental health issues like depression and anxiety to an imbalance of gut bacteria [6]. Healthy levels of healthy gut bacteria may be able to help reduce the symptoms of mental illness and potentially even prevent them from occurring in the first place. The human immune system relies heavily on good gut health to function properly. Studies have shown that having diverse microbes in your digestive tract can lead to fewer allergies, reduced inflammation, and improved immunity from environmental factors and other illnesses. Overall wellbeing: having an abundance of beneficial microbes in the digestive tract leads to an overall enhanced feeling of wellbeing due to their ability to help produce vitamins and minerals essential for bodily functions [7].

## 2. The Impact of Gut Microbiota on Choline Metabolism

Choline is fully recognized as an essential nutrient that plays crucial role in the human body, e.g., production of neurotransmitters, lipid transport, cell-membrane signaling, and methyl-group metabolism. Additionally, choline is needed in essential components for all membranes because it produces phospholipid phosphatidylcholine, lysophosphatidyl choline, choline plasmogen, and sphingomyelin. A sufficient consumption of choline is considered helpful for various illnesses, such as muscle degeneration and fatty liver [8].

Additionally, the neuroprotective properties of choline have been well-discussed thus far, to the point where it is believed that low levels of choline contribute to memory development and neural tube defects [9]. Furthermore, it is claimed that adequate dietary choline intake in adults may affect cognition, and high choline intake during pregnancy and early post-natal development might also prevent age-related memory decline, protecting the brain for Alzheimer’s disease (AD), neurological damages caused by epilepsy, and some genetic predispositions like Rett syndrome (RTT) and Down syndrome (DS) via modifications in histone and DNA methylations [10].

When converted to betaine and S-adenosilmethionine, choline may serve as a methyl-group donor and is a common precursor for membrane phospholipids and acetylcholine [11]. Humans could synthesize choline by methylation of phosphatidylethanolamine (PE) to phosphatidylcholine (PC) in their liver. In addition, choline must be obtained from dietary supplements by endogenously de novo synthesis [12].

Free choline is obtained by sodium-independent-carrier-mediated transport in the small intestine, and, on the other hand, phosphatidylcholine is created via several enzymatic processes underwent by pancreatic phospholipase A2IB and jejunal brush-border phospholipase B. It is partly recycled to phosphatidyl choline to be incorporated into chylomicrons, and a small amount of the remainder is hydrolyzed to form glycerophosphocholine. These several actions suggest that free choline, but not phosphatidylcholine, is a more available substrate for the gut microbiota’s formation of trimethylamine (TMA) [13]. Additionally, choline could also be catalyzed by choline acyltransferase into acetylcholine, which is an irreplaceable neurotransmitter in the human body [14].

Choline serves as a precursor of gut-microbiota-generated TMA, which subsequently undergoes oxidation by hepatic enzyme –Flavin monooxygenase-3 (FMO3) to form Trimethylamine-N-Oxide (TMAO) [13]. Then, TMAO enters the systemic circulation with a fasting-plasma concentration of between 2 and 40 µM in humans before urinary excretion [15]. Gut microbiota metabolizes some nutrients, including choline and phosphatidyl choline as substrates and produces TMAO. TMAO, which was once thought to be a product waste with no apparent use, is a bioactive molecule formed by the gut microbiota that several studies now indicate may play a significant role in the pathophysiology of various disorders [16,17,18].

However, it is argued that the dietary methylamine or, in the other words—TMAO—could modulate cognitive function and blood–brain barrier (BBB) integrity in vivo [15]. It is worth mentioning that, in spite of numerous studies having been conducted on this issue, further studies should be performed in order to clear the controversies. Interestingly, TMAO causes aggregation of amyloid-beta peptide and tau protein, which is the main associated pathology in AD [19]. To confirm, TMAO has the ability to cross the BBB, and TMAO found in the cerebrospinal fluid is associated with impaired cognition in AD patients [20]. Many studies show that TMAO is linked to AD, cardiovascular disease, age-related cognitive decline, and other conditions [21]. The gut microbiota affects choline metabolism in some aspects. As previously stated, certain trimethylamine-producing bacteria may reduce the availability of choline [22]. Firstly, the circulating levels of TMAO is directly associated with the amounts of dietary choline. Then, to generate TMAO, choline will decompose by a series of biological reactions to split the carbon–nitrogen bond of choline. The gut microbiota regulates these reactions, especially via Phylum *Firmicutes*, Phylum *Proteobacteria,* and six other microbial genera, including *Anaerococcus hydrogenalis*, *Clostridium asparagiforme*, *Clostridium hathewayi*, *Clostridium sporogenes*, *Escherichia fergusonii*, *Proteuus penneri*, *Providencia rettgeri,* and *Edwardsiella tarda* [16].

Yet, several investigations suggested that bacteria like *Streptococcus sanguinis* might potentially produce TMA in the mouth. There are some genes presented in Cluster C and Cluster D bacteria that are absolutely necessary for *Desulfovibrio alaskensis* and *Desulfovibrio desulfuricans* to generate TMA from choline [23]. Different hosts support each of these many plant species. Indirectly, choline metabolism is influenced by the microbiota in a significant way, by host genotype [14]. Secondly, diet is widely responsible for TMA levels. Vegetarians present different gut microbiota and decreased FMO3 activity compared with omnivorous people. In addition, dietary protein levels are directly associated with TMAO levels, and high amounts of dietary protein will lead to a higher urinary excretion. It is worth mentioning that renal clearance has a key role in order to excrete TMAO, and some believe that TMAO could take place as a glomerular injury predictor [23].

## 3. Diet Impact on Microbiota-Choline Metabolism

Choline is a nutrient that plays a vital role in brain development and the growth of the fetal brain [24]. In addition to being absorbed from food intake, this substance is also produced in a small amount in the body through the hepatic phosphatidylethanolamine N-methyltransferase pathway. Foods like eggs, fish, grains, meat, milk, and some vegetables are rich in choline, for which microbial choline TMA-lyases, including *Anaerococcus hydrogenalis*, *Clostridium asparagiformis*, *Clostridium hathewayi*, *Clostridium sporogenes*, *Desulfovibrio desulfuricans*, *Escherichia fergusoni*, *Edwardsiella tarda*, *Klebsiella pneumoniae*, *Proteus penneri*, and *Providencia rettgeri,* play a role in their absorption and conversion into TMA [25,26]. In the liver, TMA is changed into TMAO. There is convincing evidence to support the association between TMAO levels and certain diseases, including compromised renal function, colorectal cancer, and cardiovascular disease (CVD) [27]. Choline as a micronutrient is found in two forms, water-soluble and lipid-soluble, and the ways of its metabolism are also different. While the water-soluble form enters the portal circulation and reaches the liver, the lipid-soluble form enters through the lymphatic circulation [25]. An overview of choline’s effects on the nervous system shown in Figure 1.

The recommended daily intake of choline, which is typically 7.5 mg per kilogram of body weight, is influenced by factors such as age, gender, environment, and intestinal microbiota choline consumption. This amount increases during pregnancy and lactation [28]. Long-term choline deficiency is associated with many diseases, including fatty liver disease, muscle damage, hypertension, and neurological disorders. A group of mice were fed a methionine–choline-deficient (MCD) diet in a study by Trautwein et al. [29] The MCD diet also resulted in intestinal dysbiosis in addition to steatohepatitis. According to Fodor et al., choline depletion affected the levels of *Gammaproteobacteria* and *Erysipelotrichi*, which were associated with changes in liver fat [29]. The results of a study conducted by Velazquez et al. [30] suggest that dietary choline deficiency exacerbates the symptoms of Alzheimer’s disease hallmarks. Furthermore, taking less than 100 mg of choline per day have been shown to increase the risk of dementia and Alzheimer’s disease [30].

Throughout Western countries, over-the-counter choline-containing supplements are prescribed for the improvement of heart, brain, and liver health, as well as for the development of the fetal nervous system during pregnancy [31]. Some studies have demonstrated that choline supplements improve cognitive function in adults, while others have not demonstrated this effect. A study by McCann et al. [32] suggests that choline supplementation during development can enhance cognitive performance, as well as brain function changes, mainly in the hippocampus and cholinergic system. Furthermore, a placebo-controlled double-blind study demonstrated that human visuomotor performance and pupil constriction were improved after taking a choline supplement [32]. A Velázquez study found that choline supplementation may improve Alzheimer’s disease by reducing insulin resistance and microglial activation in the brain. However, a systematic review of the relationship between choline levels and cognition in adults indicates that choline supplements did not significantly enhance cognitive performance [33].

Today’s diets vary by region, culture, and socioeconomic class and, by altering the composition of the gut microbiome, have various consequences on people’s health. The Western diet, which is more common in high-income countries, includes high consumption of red meat, eggs, potatoes, and corn and low consumption of fiber. The Western diet can lead to Alzheimer’s disease through obesity, metabolic disorders, and gut microbiota dysbiosis [34]. It has been documented that the *Bacteroides* enterotype increases in Western diets, while the *Prevotella* enterotype decreases. In a study conducted on 25 people with dementia due to AD, it was shown that the level of *Bacteroides* increases. Eggs are a rich source of choline, which is also part of the Western diet (1 large egg with yolk; 147 mg). Choline is a precursor to TMAO, which increases the risk of disease, especially CVD. Zhu et al. [35] reported that taking choline and TMAO supplements increased plasma levels of TMAO in mice. According to research, individuals with less diverse gut microbiota and a greater ratio of *Firmicutes* to *Bacteroidetes* had higher TMAO responses to egg consumption [35]. A Mediterranean diet is a second diet option, which consists of a high proportion of fiber, moderate amounts of fish, dairy, and a low amount of red meat. There was a lower risk of AD and Parkinson’s disease development associated with the Mediterranean diet, according to a systematic review study. As a result of comparing the gut microbiota between MeDi and AD, the following bacteria were increased with MeDi and decreased with AD: *Lachnoclostridium*, *Slackia*, and *Parabacteroides distasonis* [36].

## 4. The Physiological Functions of Choline

Choline seems to be a water-soluble quaternary amine of the vitamin B group that the Food and Nutrition Board of the Institute of Medicine considers to be an important nutrient [37]. Choline-endogenous biosynthesis from the amino acid methionine is insufficient to meet human choline needs; thus, it is vital to consume an adequate amount of choline from fish, eggs, and meats [38]. It is a crucial precursor that facilitates the synthesis of a molecule that plays crucial functions in fetal development, specifically brain development [39]. Choline is essential in the biosynthesis of acetylcholine, phospholipids, and betaine [40]. Furthermore, choline is a precursor of the trimethylamine N-oxide metabolites produced by gut bacteria [41]. Betaine is a bioactive molecule that can preserve some of choline’s activities and is present in wheat bran, wheat germ, and spinach, making it an advantageous meal for vegetarian or vegan pregnant women [42,43].

Choline (2-hydroxyethyl-trimethyl-ammonium) is composed of three methyl groups connected to the ethanolamine nitrogen atom. Choline is an important methyl donor in betainemethionine-pathway-mediated methylation processes. Choline is a precursor of phospholipids in the human brain, including phosphatidylcholine, phosphatidylethanolamine, and sphingomyelin [44]. These choline-containing phospholipids have vital physiological roles, such as membrane formation, cellular signaling, nerves myelination, cellular division, and lipid transportation, all of which are necessary for the healthy development and regular function of the brain. Choline is also an acetylcholine (Ach) neurotransmitter precursor. Ach is the primary neurotransmitter that is produced by cholinergic neurons to control cholinergic signaling or neurotransmission [45], a mechanism that is often dysregulated in neurodegenerative disorders like Alzheimer’s disease and is marked by cognitive deficits [46]. Ach is also involved in synaptogenesis and neurogenesis. Consequently, choline-containing drugs might offer a safe route to therapy and should be further researched as feasible possibilities for the early-stage treatment of neurodegenerative illnesses. While choline is acquired from the food, many individuals do not meet the optimum daily consumption needs of 7.5 mg per kg of body weight. This ideal value has been demonstrated to differ across people based on variables like gender, age, genetic variations, and environment [47].

After consuming meals high in choline, plasma choline levels increase. Choline in circulation may pass the blood–brain barrier (BBB) and be further metabolized in the brain, or it can be mostly oxidized in the liver to betaine to serve as a key source of methyl groups in this organ. Choline may also be generated de novo from phosphatidylcholine (PC) via a series of complicated chemical events that have been found to occur largely in the liver and to a lesser degree in other tissues, such as the brain. In mammals, phosphatidylethanolamine-N-methyltransferase (PEMT) catalyzes the de novo synthesis of choline by the successive methylation of phosphatidylethanolamine (PEM) to create PC, utilizing SAM as a methyl donor [48]. Choline’s potential benefits and drawbacks on the nervous system shown in Figure 2.

By means of the enzymatic activity of choline oxidase, a substantial portion of the dietary choline that enters the liver may be irreversibly transformed into betaine. This oxidation is a source of methyl groups for the production of methionine and SAM, two of the most important methyl donors for methylation pathways. With the Ach-synthesizing enzyme choline acetyltransferase, a tiny percentage of dietary choline is typically acetylated to Acetyl-CoA to produce Ach (ChAT). PEMT and ChAT activity were detected in cholinergic nerve terminals, indicating that these terminals are capable of synthesizing choline for use in Ach production in neurons [49].

Choline is shown to enter the brain from the systemic circulation through transporters or transport proteins situated in the BBB. This means that variations in choline levels in the blood might influence choline levels in the brain and the amount of choline used by neurons for the creation of Ach. In addition, research demonstrates that choline may be produced from scratch in the soma of neuronal cells and in presynaptic terminals [50], finding another supply of choline for neuron structural and functioning needs. The Kennedy cycle involves numerous phases in the endogenous production of PC, a cellular membrane structural component. Choline kinase (CK) phosphorylates choline to generate phosphocholine. Phosphocholine with CTP create CDP-choline. CDP-choline: 1, 2, diacylglycerol cholinephosphotransferase then forms PC from DAG and CDP-choline (CPT). As mentioned, membrane PC choline can be employed for Ach biosynthesis. Acetylcholinesterase degrades Ach produced by presynaptic cholinergic neurons into choline and acetic acid in the synaptic cleft (AchE). This breakdown releases choline in the synaptic cleft, which presynaptic neurons take up via choline transporters (mainly CHT1). Cholinergic neurons employ this transporter-mediated reuptake to synthesize Ach or phosphorylate membrane PC [51]. To summarize, free choline that crosses the BBB from systemic circulation, choline liberated from PC by phospholipases enzymes, and choline picked up from the synaptic cleft by choline transporters (CHTs) will be reutilized in membrane PC after Ach degradation by AchE and are the main sources of choline in the brain. Hence, PC and Ach store free choline in the brain, while ChAT and AchE measure cholinergic neuronal activity or neurotransmission. Choline shortage has been associated with numerous illnesses or disorders due to its wide-ranging effects on metabolism, suggesting that maintaining an appropriate amount of this vitamin between early childhood and maturity may improve physical and mental health. Animal and human research demonstrated that loss of choline from the diet during adulthood might produce organ dysfunctions such fatty liver and liver damage owing to homocysteine increase [52,53]. In humans, choline deprivation caused lymphocyte apoptosis, muscle injury, and DNA destruction [54]. As estrogen synthesizes choline, premenopausal women need less choline from food than postmenopausal women. Estrogen binds to estrogen-responsive elements (EREs) along the PEMT gene to activate its expression for choline synthesis [55]. Hence, post-menopausal women may lose choline synthesis due to estrogen depletion or genetic polymorphism or SNPs in the hepatic PEMT gene, which may raise their risk of illnesses or organ damage like cancer [56]. Choline deficiency during prenatal, perinatal, or postnatal life may cause neural tube abnormalities, mental deterioration with age [57], and neurological problems [58].

CDP-choline, another choline-containing molecule, has been demonstrated to prevent memory loss or deterioration in old rats when fed in their diets for an extended length of time. Inferred from their performance in memory-related activities, such as the Morris water maze, these elderly rats displayed a slower rate of memory impairment over time as a consequence of this long-term feeding [59]. The favorable implications of CDP-choline usage in an aging human population have not yet been investigated, although additional research may be warranted. It has been widely known that a reduction in membrane PC and PEM levels contributes to neurodegeneration [60].

Alzheimer’s disease is a progressive, devastating, neurodegenerative ailment that is characterized by the gradual degradation of cholinergic neurons and a loss in cholinergic neurotransmission in the aging brain [61]. Since choline contributes to the structural stability of membranes and cholinergic neurotransmission, it would be helpful to think about the possibility that choline supplementation in the daily intake in combination with other therapeutic interventions could increase brain activity and possibly delay memory and cognitive functions or mitigate some negative behavioral effects that are commonly associated with aging. The regulation of cellular membrane integrity has been the subject of much research and has been considered essential for cellular processes, including neuronal membrane functions such as cell signaling and neurotransmission. Choline is a precursor of both phosphocholine (PC) and sphingomyelin (SM), two structural components of membranes. SM promotes several key physiological activities, including the myelination of nerves, a process that is often amplified during fetal development and crucial for neuronal communication in the adult brain. It is thus not unexpected that excessive amounts of choline in the brain would impair the availability of this structural component and might possibly have negative effects on brain circuitry and change brain function throughout the course of the individual’s lifetime [62].

## 5. Gut Microbiota, Choline Metabolites, and Neurodegenerative Diseases

Neurodegenerative disorders have become a health issue currently, and the prevalence of Alzheimer, Parkinson, and Huntington disease is skyrocketing as people age [63,64]. The aforementioned disorders usually cause impairments in learning, memory, and cognitive functions. Also, they result in sleep disturbance, behavioral change, mood swings, and lack of productivity that can leave an enormous burden on affected individuals and the society [65]. Studies showed that in addition to genetic factors, environmental parameters such as long term exposure to toxins, stress, and some nutrients are responsible for neurodegenerative disorders, for instance, some diets enriched with Vit B12, Vit B6, choline, betaine, and folic acid can mitigate decline in cognitive functions and positively affect many genes’ expressions during crucial periods like prenatal, perinatal, early postnatal, or during adulthood [9,66]. Fetal life is an essential period for brain development. Although cellular division, neurogenesis, etc., are programmed processes, the brain is flexible enough to change under some circumstances during this critical time, for instance, deficiency in choline intake during pregnancy can influence the offspring’s cognitive functions in the future, including causing neural tube defects in humans and rodents, decrease in neural progenitor cells proliferations, and impairments in memory-related tasks [63].

Choline and some other nutrients like omega-3 fatty acid and uridine have positive effects on early AD symptoms and help patients with boosting their memory and enhancing their neuronal connectivity. For starters, choline is a necessity for biosynthesis of acetylcholine (Ach), a required neurotransmitter in cholinergic pathways and a significant item in synapse establishment. Also, it is a structural element in phosphatidylcholine (PC) and sphingomyelin (SM) formation in cell membrane; thus, the integrity of cellular membrane and other various functions like cellular signaling, nerve cell myelination, cellular division, etc., are maintained through choline [29,66]. Choline and its derivatives, like betaine, participate in methylation pathways too. Any changes in the natural process of DNA and histone methylation can alter gene expression in a way that either silences or expresses genes [29]. Choline supplementation in the prenatal period alters memory-related gene expression through some molecular changes in the hippocampus and the cerebral cortex, like insulin-like growth factor-1 (IGF-1), IGF-II, and its receptor. Hypomethylation in an amyloid precursor transmembrane protein (APP) gene promotor or some changes in histone levels have been detected as molecular changes in AD patients [9].

Another example of altered methylation happens in stress-related genes like corticotropin-releasing hormone genes via maternal choline intake. This molecular change results in less stress hormone levels in the cord blood, and as has been pointed out before, stress is a risk factor in neurodegenerative disorders. Also, choline improves neurogenesis and increases brain-derived neutrophic factors (BDNFs) [67,68]. In addition to its molecular and behavioral effects, prenatal choline supplementation increases the size of some cholinergic neurons’ cell bodies, raises their acetylcholine production level, lowers their acetylcholine esterase activity, develops neuronal linkage in the hippocampus, and raises their levels of N-methyl-D-aspartate (NMDA) neurotransmission and nerve growth factor (NGF). Furthermore, to an aberrant membrane structure, decreased Ach production and impaired NMDA receptor-mediated signaling are two probable causes of Alzheimer disease [63].

Studies have shown a decrease in membrane phospholipids with aging, and since choline is required for the production of phospholipids and its level decreases with age, we can draw the conclusion that cellular membrane integrity, its viability and ability to function properly, and our ability to perform memory-related tasks are all threatened in neurodegenerative disorders like ALS as the years pass and choline levels decline [68]. Choline and its metabolites in the gut microbiota can also modify the immune system. In reality, because it regulates the creation of several neurotransmitters and hormones including dopamine, adrenaline, serotonin, and GABA, the gut microbiota have a considerable influence on the neurological system. These are all components of the enteric or central nervous system [64,69].

Also, the gut microbiota regulates many immune-signaling pathways, like the nuclear factor NF-KB-signaling pathway. Research revealed a correlation between elevated NF-KB levels and the expression of TNF- and the symptom of forgetfulness. In conclusion, it is inevitable that the gut microbiota’s changes will have an impact on the immune system, which will then have an impact on the central nervous system [64]. Consequently, choline has essentially remarkable functions in various cellular pathways, and studies indicate that choline functions and acetylcholine neurotransmission are dysregulated in many neurodegenerative disorders like in Alzheimer disease. Acetylcholine esterase (AchE) and choline acetyltransferase (ChAT) are not produced as normal and cholinergic neurons in the basal forebrain, which are highly responsible for learning, memory, attention and cognitive functions and can modulate the degeneration of cortical neurons [9,63]. Furthermore, a choline-rich diet during the prenatal and perinatal period attenuated the symptoms of schizophrenia, Down’s syndrome, Rett syndrome, and Alzheimer disease [9]. Choline intake can develop short-term memory in both animal cases and human samples.

The fact that choline levels are linked to and fluctuate in a wide range of diseases, including hypoxia, seizures, and others, should also be highlighted. This matters since conditions like seizure and ischemia can impose long-term unfortunate effects on brain function and facilitate the progression of neurodegenerative disorders [63].

## 6. Potential Therapies in Choline-Related Diseases

Currently, the incidence of neurodegenerative diseases (NDs) is increasing due to advances in the age of the global population. Therefore, in order to increase the quality of life and a reduction in healthcare costs for these patient, effective treatment options must be available to prevent, treat, or manage them. Unfortunately, there have been no major medical treatment breakthroughs that effectively stop the progression of neurodegenerative diseases. Due to their nature, which is the dysfunction of nervous systems and their neurotransmitters, various studies have been conducted in the past decades to find a novel approach to managing treatments [70,71].

Nutrition has progressively been identified as an essential variable impacting the vulnerability to and incidence of neurodegenerative diseases. To our knowledge, choline is a necessary nutrient and has a potential role in numerous nervous system functions, including attention, learning, cognition, and memory. Choline is required to sustain nervous system function through cholinergic and phospholipid metabolism and synaptic membrane formation. Recently, a few animal and human studies have been conducted to evaluate the effect of choline supplementation therapy on patients with neurodegenerative diseases [66].

Dave N., et al. [67] assayed 3xTg-AD, a model of AD, and non-transgenic (NonTg) control mice on either a standard laboratory diet with sufficient choline (2.0 g/kg choline bitartrate) or a choline-deficient diet (no choline bitartrate) from 3 to 12 months. Analysis of heart and liver tissues revealed an increase in the soluble and insoluble structure of amyloid-β and thioflavin S. In addition, tau hyper-phosphorylation was increased at different pathological epitopes in the hippocampus and cortex of 3xTg-AD mice. Dietary choline deficiency impairs the hippocampal network involved in microtubule function and postsynaptic membrane regulation. Also, this deficiency altered the plasma protein network related to insulin metabolism, mitochondrial function, inflammation, and fructose metabolism. Lasting dietary choline intake is imperative to prevent and reduce the pathological features of Alzheimer’s disease [67]. 

The study looked at how choline supplementation affected the APP/PS1 mice model of Alzheimer’s disease (AD) in terms of reducing memory loss and pathology similar to Alzheimer’s disease (AD). They treated female APP/PS1 and non-transgenic (NonTg) mice with either a control choline (1.1 g/kg choline chloride) or a choline-supplemented diet (5.0 g/kg choline chloride) from 2.5 to 10 months. Then, the mice were tested in a Morris water maze to assess spatial memory and neuropathological evaluation. During this study, choline supplementation significantly reduced amyloid-β plaque load and developed spatial memory in APP/PS1 mice. These changes are associated with decreased amyloidogenic processing of APP, reduced activation of disease-associated microglial, and downregulation of α7nAch and σ1 receptors. Lifelong dietary choline supplementation may reduce disease burden and the development of cognitive deficits. Thus, the benefit of choline supplementation on reducing microglia activation in non-Tg mice demonstrates a benefit to the aging brain [66].

The study by Wang Y. tested APP/PS1 mice and their wild-type littermates using control or choline diets from 2 to 11 months of age. Compared with wild-type mice, control-diet APP/PS1 mice are characterized by decreased forebrain cholinergic neurons, decreased intensity of cholinergic fiber staining in the amygdala, and decreased choline levels in the hippocampus and cerebral cortex. Choline levels partially prevented these changes and ameliorated cognitive deficits and anxiety. In addition, the choline diet on APP/PS1 mice reduced amyloid-β deposition and microgliosis. These effects may be due to the inhibition of NLRP3 (nucleotide-binding domain, leucine-rich-containing family, pyrin-domain-containing-3) inflammasome activation and restoration of synaptic membrane formation. In the AD pathophysiology, cholinergic degeneration has an important role, and choline can reduce this degeneration [72].

Alldred M., et al. [73] evaluated the effects of choline supplementation on a mouse model of AD. Choline reduces spatial awareness and attentional dysfunction and has protective effects on the survival of basal forebrain cholinergic neurons in Ts65Dn mouse models. The study indicated that choline normalized the expression of several genes, including synaptic plasticity, calcium signaling, and AD-related neurodegeneration associated with beta-amyloid peptide clearance. Choline is a possible treatment for long-term reprogramming of dysregulated neural signaling in the brains of AD mice. Choline is a necessary nutrient for normative and improving the function of brain pathways [73]. A lower risk of dementia and AD was linked to higher dietary choline consumption, respectively. Patients without dementia were followed for 16.1 ± 5.1 years (1991–2011) and screened by a valid self-reported Harvard food frequency questionnaire (FFQ). Total choline and its contributing compounds were estimated based on the published nutritional data. The link between dietary choline intake and dementia and AD was examined, and it found that there were 247 cases of dementia, of which 177 were AD. Dietary choline intake shows a non-linear relationship between dementia and AD. After adjusting for covariates, low choline intake (≤219 mg/day for dementia and ≤215 mg/day for AD, respectively) was significantly associated with incident dementia and AD. Therefore, this study suggests that essential nutrients such as choline supplementation potentially influence brain development, disease risk, and prevention [30].

A lower risk of incident dementia was associated with a higher choline intake. They evaluated the association between a regime of choline supplements and the risk of dementia in middle-aged and older men in the prospective study. Dementia-free patients between the ages of 42 and 60 were examined, and 482 patients completed five cognitive performance tests four years later. Those diagnosed with dementia and Alzheimer’s disease were extracted from the Finnish health register. This study revealed that the mean total choline intake was 431 ± 88 mg/day. During a follow-up of 21.9 years, 337 men were diagnosed with dementia. The patients with the highest versus lowest choline consumption had a 28% lower risk of incident dementia. Total choline intake was not associated with incidents of dementia risk. However, it was associated with better performance on cognitive tests measuring frontal and temporal lobe function and better performance in memory function and fluency [74].

The circulating level of choline is lower in patients with Alzheimer’s disease (AD) than in healthy individuals. This study investigated to what extent the level of the micronutrient choline might be affected by oral supplementation, using data obtained from three randomized clinical trials and an open-label extension (OLE) study with 12 to 48-week follow-up data. Patients with mild or mild-to-moderate AD received active or controlled products once daily for 12–24 weeks or active products (contains 400mg choline along with other nutrients) during the 24 weeks. The plasma level of choline was measured and significantly increased in the active versus control group from baseline to 24 weeks in all AD types (all *p* < 0.001). During the OLE study, the plasma level of choline was significantly increased in the control-active group from week 24 to week 48, after which the control group switched to using the active product. During the OLE study, it remained consistently elevated in the active-active group (they continued to use active product). This study suggests that choline concentration, known to decrease in the AD population, increased after 24 weeks of daily use of the active product compared with a control product in patients with mild and mild-to-moderate AD [75]. According to clinical research conducted on both humans and animals, early NDs can benefit from a combination of choline supplements and the drugs currently used to treat NDs in order to improve memory performance and synaptic functioning. Clinical trials have demonstrated that choline supplements have a good effect on the progression of the disease as well as lowering the risk of having these illnesses. Consequently, we suggest using more animal and human samples to research long-term follow-up of choline treatments in early and prodromal NDs [76,77,78]. 

## 7. Conclusions

Choline is thought to have neuroprotective qualities and is necessary for the production of neurotransmitters, lipid transport, cell-membrane signaling, and methyl-group metabolism. Chronic choline insufficiency is linked to a number of disorders, and supplements are recommended for better health. Acetylcholine (Ach) neurotransmitter precursor and phospholipid precursor choline can be synthesized de novo from phosphatidylcholine (PC) by the enzymes phosphatidylethanolamine-N-methyltransferase (PEMT) and choline oxidase. Choline has been found to improve amyloid-β, thioflavin S, and tau hyper-phosphorylation in patients with neurodegenerative disorders like Alzheimer’s disease. A lower risk of dementia and AD was associated with higher dietary choline consumption, respectively. Also, choline is necessary for normal brain function and can enhance the efficiency of neural networks while reducing amyloid accumulation and microgliosis. Clinical research suggests that early NDs may benefit from a combination of choline supplements and the drugs currently used to treat NDs in order to improve memory and synaptic function.

## Figures and Tables

**Figure 1 biomolecules-14-01345-f001:**
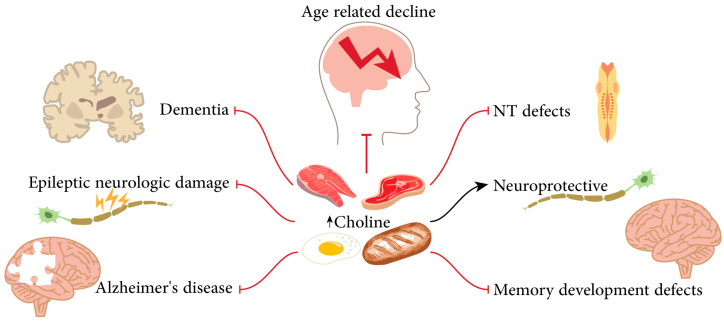
An overview of choline’s effects on the nervous system. Taking sufficient dietary choline may affect cognition, and a high-choline diet in early life might also prevent age-related memory decline, and protect the brain from Alzheimer’s disease (AD), and neurological damage caused by epilepsy. Furthermore, sufficient choline prevents neural tube defects, dementia progression, and has neuroprotective effects.

**Figure 2 biomolecules-14-01345-f002:**
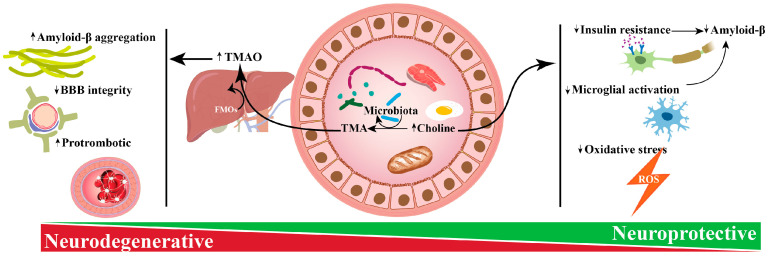
Choline’s potential benefits and drawbacks on the nervous system. Research indicates that choline has a varied impact on the nervous system. Dietary choline decreases insulin resistance in the nervous system. Additionally, choline lessens microglial activation, reducing the buildup of amyloid beta in the brain—a hallmark of Alzheimer’s disease. Besides, choline has been linked to a reduction in oxidative stress, associated with many neurodegenerative disorders including Alzheimer’s disease, Parkinson’s disease, Huntington’s disease, multiple sclerosis, and amyotrophic lateral sclerosis (ALS). On the other hand, through the microbiota, choline is converted to trimethylamine (TMA). In the liver, TMA converts to Trimethylamine-N-Oxide by flavin monooxygenase enzymes. This increase in TMAO leads to increased amyloid beta aggregation, interrupted BBB integrity, and increased prothrombotic effects.

## Data Availability

The corresponding author will provide the datasets created during and/or analyzed during the current investigation upon reasonable request.

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
