# Peer review of "The Importance of Gut Microbiota on Choline Metabolism in Neurodegenerative Diseases"

_biomolecules, 2024, doi:10.3390/biom14111345_

Round 1

Reviewer 1 Report

Comments and Suggestions for Authors

This is a well-written, timely, and comprehensive narrative review that highlights the important role of choline in human health, including its epigenetic effects and the influence of microbiota on choline production and metabolism. From a scientific perspective, the content of this manuscript is sound, and the implications of the presented data are well justified. There are some areas that with minor modifications the quality of this work can be improved:

-            In some areas, the transitions between concepts could be improved. For instance, the paragraph corresponding to lines 472-483 could begin as follows: “In another study, a lower risk of incident dementia was associated with higher choline intake. The investigators evaluated...”. Similar adjustments could enhance the flow of the text in the following paragraph (484-).

-            Under the subtitle “Gut Microbiota, Choline Metabolites, and neurodegenerative diseases”, the authors have not provided sufficient data on the gut microbiota. It would be reasonable to either include more information about the gut microbiota and its relationship with neurodegenerative diseases or modify the subtitle accordingly. For example, they could remove “gut microbiota' from the subtitle, as the link between gut microbiota and choline metabolism is already sufficiently discussed under the subtitle 'Diet Impact on Microbiota-Choline Metabolism".

-            In certain sections of the main text, the authors repeat the abbreviated names of specific chemicals. For example, 'phosphatidylcholine (PC) and sphingomyelin (SM) formation' is redefined in lines 351-352, despite being introduced earlier. For consistency, each chemical or compound should be defined or summarized only once. Subsequently, the abbreviations PC and SM can be used instead of repeating the full names.

-            Finally, perhaps, adding another figure to illustrate the chemical structures and details of the reactions that generate choline derivatives or metabolites (e.g., betaine, TMA, TMAO, and others) could further enhance the quality of this work.

Author Response

We thank the reviewers for the evaluation and constructive feedback.

Q1. In some areas, the transitions between concepts could be improved. For instance, the paragraph corresponding to lines 472-483 could begin as follows: “In another study, a lower risk of incident dementia was associated with higher choline intake. The investigators evaluated...”. Similar adjustments could enhance the flow of the text in the following paragraph (484-).

Answer1. The manuscript was revised accordingly and the changes are highlighted in the manuscript.

Q2. Under the subtitle “Gut Microbiota, Choline Metabolites, and neurodegenerative diseases”, the authors have not provided sufficient data on the gut microbiota. It would be reasonable to either include more information about the gut microbiota and its relationship with neurodegenerative diseases or modify the subtitle accordingly. For example, they could remove “gut microbiota' from the subtitle, as the link between gut microbiota and choline metabolism is already sufficiently discussed under the subtitle 'Diet Impact on Microbiota-Choline Metabolism".

Answer2. The manuscript was revised accordingly and the changes are highlighted in the manuscript.

Q3. In certain sections of the main text, the authors repeat the abbreviated names of specific chemicals. For example, 'phosphatidylcholine (PC) and sphingomyelin (SM) formation' is redefined in lines 351-352, despite being introduced earlier. For consistency, each chemical or compound should be defined or summarized only once. Subsequently, the abbreviations PC and SM can be used instead of repeating the full names.

Answer3. It was corrected.

Q4. Finally, perhaps, adding another figure to illustrate the chemical structures and details of the reactions that generate choline derivatives or metabolites (e.g., betaine, TMA, TMAO, and others) could further enhance the quality of this work.

Answer4. A new figure was designed and added, i.e. Figure 3

We thank again the Reviewer for the positive evaluation

Reviewer 2 Report

Comments and Suggestions for Authors

Abstract:

Refine the abstract by focusing on aims and objectives of the review, key outcomes, condensing background information, and making it more accessible to a broader audience.

Introduction:

The introduction provides a solid foundation for the topic, highlighting the significance of gut microbiota and choline metabolism. However, it could better emphasize the research gap being addressed. The rationale for why this review is needed should be stated more explicitly.

Methods:

If the review includes any specific methodologies for literature selection or analysis, it would be useful to include this for transparency.

Results:

The results of studies discussed should be critically analyzed rather than simply described. For example, some statements about choline supplementation improving cognitive performance are followed by contradictory studies, but this is not well explored in the discussion.

Include more critical analysis of conflicting findings in the reviewed literature, highlighting potential reasons for discrepancies (e.g., differences in study design, population, or dosages).

Conclusion:

The conclusion provides a good summary but should offer a clearer statement on future research directions and practical implications of the findings.

Add a few lines on potential clinical applications of choline supplementation and what future studies should focus on.

3. Tables and Figures:

A comparative tabular analysis should be encouraged.

Figure 1 captions could be more descriptive to ensure clarity without needing to refer to the text. For instance, Figure 1's caption should explain the main findings in a way that can stand alone.

Comments on the Quality of English Language

Some sentences are lengthy and contain multiple ideas that can be broken down for clarity. For example, in the abstract, combining several ideas into one sentence without clear segmentation might make it harder for readers to follow.

The manuscript contains several grammatical errors, such as inconsistent use of articles, verb tenses, and awkward phrasing. For instance, phrases like "can develop short-term memory in both animal cases and human samples" could be reworded as "can improve short-term memory in both animal models and human subjects."

Author Response

We thank the reviewer for the evaluation of our work and for the constructive feedback.

Abstract:

Q1. Refine the abstract by focusing on aims and objectives of the review, key outcomes, condensing background information, and making it more accessible to a broader audience.

Answer1. The abstract has been revised accordingly

Introduction:

Q2. The introduction provides a solid foundation for the topic, highlighting the significance of gut microbiota and choline metabolism. However, it could better emphasize the research gap being addressed. The rationale for why this review is needed should be stated more explicitly.

Answer2. The Introduction has be updated accordingly, the changes are marked in the text.

Despite gut microbiota and its involvement in general health have been extensively studied, little is known about how gut microbiota particularly affects choline metabolism in connection to neurodegenerative disorders. Although choline is recognized to be crucial for brain function, neurodegenerative diseases like Alzheimer's disease (AD) are increasingly being connected to the gut microbiota's conversion of choline into bioactive molecules like trimethylamine-N-oxide (TMAO). However, it is still unknown how precisely this metabolic interaction leads to neurodegeneration. Furthermore, the significance of the gut microbiota in regulating choline's effects on cognitive health and illness is not completely taken into account in the current dietary guidelines for choline consumption. By combining current research on how changes in the diversity of the gut microbiota impact choline metabolism and play a role in the pathogenesis of neurodegenerative disorders, this review seeks to fill in these gaps. This study aims to shed light on the intricate interactions between microbiota, neurodegeneration, and food by thoroughly examining the gut-brain axis via the lens of choline metabolism. This might lead to the development of novel treatment approaches. This emphasis is particularly relevant in light of the growing incidence of neurodegenerative illnesses and the growing interest in treatments that target the microbiota.

Methods:

Q3. If the review includes any specific methodologies for literature selection or analysis, it would be useful to include this for transparency.

Answer3. We used a methodical approach to the selection of literature to guarantee an in-depth and objective evaluation. The last ten years' peer-reviewed publications on gut microbiota, choline metabolism, and neurodegenerative disorders were sought after by searching databases including PubMed, Scopus, and Web of Science. Search terms that were employed in the process were "choline metabolism," "gut microbiota," "trimethylamine-N-oxide (TMAO)," and " Neurodegenerative diseases".

Results:

Q4. The results of studies discussed should be critically analyzed rather than simply described. For example, some statements about choline supplementation improving cognitive performance are followed by contradictory studies, but this is not well explored in the discussion.

Include more critical analysis of conflicting findings in the reviewed literature, highlighting potential reasons for discrepancies (e.g., differences in study design, population, or dosages).

Answer4. According to the reviewer's recommendation, the manuscript was updated and the changes were added and highlighted at the end of the discussion of the article.

There are notable distinctions even though choline supplementation might reduce the progression of neurological disorders and improve cognitive function. For example, in animal models of AD, choline supplementation has been shown to have a considerable positive impact on memory and synaptic function, especially in terms of lowering the burden of amyloid-beta plaque and enhancing cognitive outcomes. Nevertheless, choline supplementation has not been found to significantly increase cognitive function in other research, including some randomized controlled trials in people. These differences might be the result of several things. First, variations in the research design and technique, including the amount of time supplemented and the type of choline given, might affect the results. For instance, although phosphatidylcholine, or CDP-choline, has a different bioavailability and potential effects on brain function, choline chloride is used in certain research while CDP-choline is the subject of others. Second, demographic changes can impact supplement response as well. These variations include age, baseline cognitive state, and genetic variations such polymorphisms in genes associated to choline metabolism (like PEMT). Furthermore, doses differ significantly throughout trials. Higher choline dosages are frequently employed in animal models, which may make them less directly similar to doses used in human studies. Moreover, individual variations in microbiome composition may result in varying quantities of bioactive metabolites such as TMAO, which might inject heterogeneity into the gut microbiota's function in influencing choline metabolism. Research that don't take these microbiota-related variables into consideration might be missing something crucial that affects how successful choline therapies are. Finally, it can be important to time the addition of choline supplements. In contrast to human research, which often start supplementing later in life, animal studies frequently start supplementation early in life or during pregnancy, which may have more significant neuroprotective benefits. This presents the issue of whether there is a window of opportunity for intervention that is optimal but has not received enough attention in human research. The conflicting findings highlight the need for more standardized research techniques, such as comparable choline supplementation forms, uniform dose, and a deeper grasp of individual diversity in gut flora and genetic variables. Long-term follow-ups should be taken into account in future research to ascertain whether the effects of choline supplementation on cognition are maintained over time or decline as neurodegeneration advances.

Conclusion:

Q5. The conclusion provides a good summary but should offer a clearer statement on future research directions and practical implications of the findings.

Add a few lines on potential clinical applications of choline supplementation and what future studies should focus on.

Answer5. In the conclusion of the article, potential clinical applications of choline supplementation and what future studies should focus on were added.

The review emphasizes the crucial role of gut microbiota in choline metabolism and its potential impact on neurodegenerative diseases like Alzheimer's. While choline supplementation may reduce amyloid-beta plaque burden and enhance cognitive performance, contradictory results suggest the need for more comprehensive studies, considering individual characteristics and supplementation forms.

Standardizing choline supplementation protocols, especially with regard to dose, time, and kind of choline administered, should be the main emphasis of future study. Clinical studies should also investigate the impact of gut microbiota and develop tailored therapies that take into account microbiome makeup and genetic variables influencing choline metabolism. To evaluate the durability of cognitive gains and identify the best times to intervene, long-term research is required.

Tables and Figures:

Q6. A comparative tabular analysis should be encouraged.

Figure 1 captions could be more descriptive to ensure clarity without needing to refer to the text. For instance, Figure 1's caption should explain the main findings in a way that can stand alone.

Answer6. The caption of Figure 1 has been revised accordingly.

In addition, we have revised the text of the manuscript to fix typos and language-related issues.

We thank the Reviewer again for the feedback